# DSP: Dynamic Semantic Prototype for Generative Zero-Shot Learning

## Abstract

Conditional generative models (*e.g.*, generative adversarial network (GAN)) have advanced zero-shot learning (ZSL). Studies on the generative ZSL methods typically produce class-specific visual features of unseen classes to mitigate the issue of lacking unseen samples based on the predefined class semantic prototypes. As these empirically predefined prototypes are not able to faithfully represent the actual semantic prototypes of visual features (*i.e.*, visual prototypes), existing methods limit their ability to synthesize visual features that accurately represent real features and prototypes. We formulate this phenomenon as a visual-semantic domain shift problem. It prevents the conditional generative models from further improving the ZSL performance. In this paper, we propose a dynamic semantic prototype learning (DSP) method to align the empirical and actual semantic prototypes for synthesizing accurate visual features. The alignment is conducted by jointly refining semantic prototypes and visual features so that the conditional generator synthesizes visual features which are close to the real ones. We utilize a visual→semantic mapping network (V2SM) to map both the synthesized and real features into the class semantic space. The V2SM benefits the generator to synthesize visual representations with rich semantics. The visual features supervise our visual-oriented semantic prototype evolving network (VOPE) where the predefined class semantic prototypes are iteratively evolved to become dynamic semantic prototypes. Such prototypes are then fed back to the generative network as conditional supervision. Finally, we enhance visual features by fusing the evolved semantic prototypes into their corresponding visual features. Our extensive experiments on three benchmark datasets show that our DSP improves existing generative ZSL methods, *e.g.*, the average improvements of the harmonic mean over four baselines (*e.g.*, CLSWGAN, f-VAEGAN, TF-VAEGAN and FREE) by 8.5%, 8.0% and 9.7% on CUB, SUN and AWA2, respectively.

## 1 Introduction

Zero-shot learning (ZSL) recognizes the unseen classes, by transferring semantic knowledge from some known classes to unknown ones. Recently, conditional generative models (*e.g.*, generative adversarial networks (GANs) Goodfellow et al. (2014) and variational autoencoder (VAE) Kingma & Welling (2014)) have been successfully applied in ZSL and achieved promising performance. They synthesize the class-specific images or visual features of unseen classes to mitigate the lack of unseen samples based on the condition of the predefined class semantic prototypes Arora et al. (2018); Xian et al. (2018; 2019b); Chen et al. (2021a;b). Because of the high quality synthesis of GAN, studies arise to generate unseen samples in the CNN features to benefit ZSL Yan et al. (2021); Xian et al. (2019b).

The class-specific unseen sample generations are conditioned on the semantic prototypes. However, there is a limitation that the empirically predefined semantic prototypes are not able to faithfully represent the actual semantic prototypes of visual features (*i.e.*, visual prototypes). As shown in Fig. 1, *i)* the predefined class semantic prototypes are annotated by human, inevitably resulting in some inaccurate annotations (*e.g.*, the attribute "bill color orange" in Sample-2 in Fig. 1(a)), and *ii)* the visual images are with multi-views, resulting in the phenomenon that some importantly annotated attributes do not appear in the visual representations (*e.g.*, the attribute "bill color orange" in Sample-3 and attribute "leg color black" in Sample-4 in Fig. 1(a)). That is, some visual representations of one

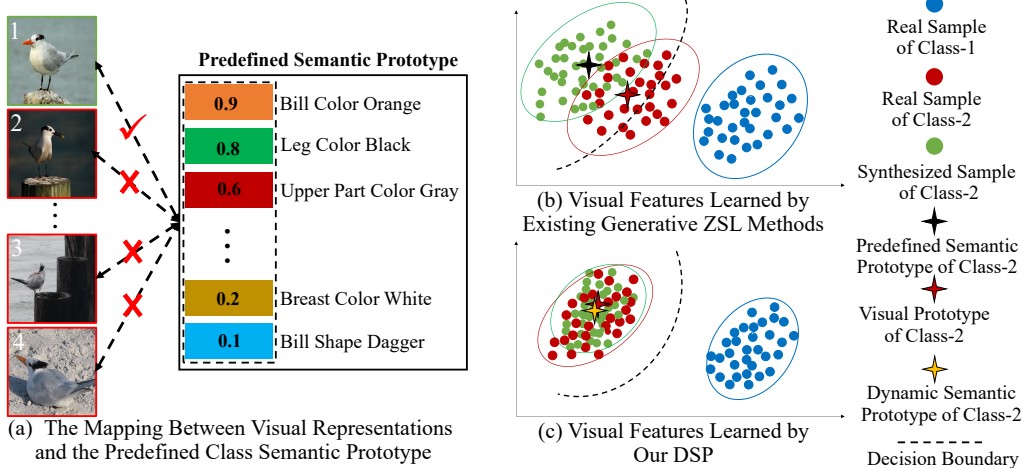

Figure 1: Motivation illustration. **(a)** The fact that some visual representations of one class are incorrectly mapped with their common predefined semantic prototype. **(b)** Existing generative ZSL methods merely generate visual features guided by the predefined class semantic vectors that fail to accurately represent the semantic prototypes of visual representations (also denoted visual prototypes), resulting in the synthesized visual features far away from the corresponding real visual features and real semantic prototypes. **(c)** Our DSP dynamically evolves the predefined semantic prototypes as dynamic semantic prototypes that are closer to their visual prototype, enabling the generator to synthesize reliable visual features and enrich semantic information into visual features with the dynamic semantic prototypes. (Best viewed in color)

class are incorrectly mapped with their common predefined semantic prototype. As such, the visual features synthesized by existing generative ZSL methods are far away from their corresponding real visual features and visual prototypes (as shown in Fig. 1(b)), which heavily limits their classification performance. We formulate this phenomenon as a problem of *Visual-Semantic Domain Shift*. Therefore, it is essential to refine the empirical defined semantic prototype. Using an accurate prototype benefits the generator supervision to improve the generative ZSL performance.

In light of these observations, we argue that the predefined semantic prototypes can be refined to become consistent to the actual semantic prototypes (*i.e.*, visual prototypes) according to the visual information. As such, the refined semantic prototypes can capture the visual samples exactly and act as an accurate conditional signal for the generator, enabling generative ZSL methods to learn a desirable semantic→visual mapping (*i.e.*, generator). Targeting this goal, in this paper, we propose a dynamic semantic prototype learning (DSP) method to align the empirical and actual semantic prototypes for synthesizing reliable visual features. The alignment is conducted by jointly refining semantic prototypes and visual features so that the conditional generator synthesizes class-specific visual features which are close to the real ones, as shown in Fig. 1(c). Accordingly, our DSP well tackles the visual-semantic domain shift problem and advances the generative ZSL methods.

Specifically, DSP consists of a visual→semantic mapping network (V2SM) and visual-oriented semantic prototype evolving network (VOPE). Cooperated with the generator, V2SM maps visual features into the class semantic space, enabling the conditional generator to synthesize class-specific visual samples with rich semantic information and convey visual information into the VOPE. Under the supervision of visual information, VOPE iteratively evolves and refines the predefined class semantic prototypes to become dynamic semantic prototypes, which are fed back to the generative network as a conditional supervision to encourage the generator for reliable visual feature synthesis. Finally, DSP concatenates the evolved semantic prototypes into visual features for enhancement, enabling the visual features to be closer to their corresponding semantic prototypes and alleviate the cross-dataset bias problem Chen et al. (2021a). Extensive experimental results demonstrate consistent performance gains over the state-of-the-art generative ZSL methods on three challenging benchmark datasets, *i.e.*, CUB Welinder et al. (2010), SUN Patterson & Hays (2012) and AWA2 Xian et al. (2019a). For example, the average improvements of harmonic mean over four baselines (*e.g.*, CLSWGAN Xian et al. (2018), f-VAEGAN Xian et al. (2019b), TF-VAEGAN Narayan et al. (2020) and FREE Chen et al. (2021a)) are **8.5%**, **8.0%** and **9.7%** on CUB, SUN and AWA2, respectively. Notably, our DSP is flexible and effective to be entailed in any generative ZSL method.

## 2 RELATED WORK

**Embedding-based Zero-Shot Learning.** Embedding-based ZSL methods were popular early, they learn a visual→semantic projection on seen classes that is further transferred to unseen classes Akata et al. (2016; 2015); Huynh & Elhamifar (2020a); Chen et al. (2022); Huynh & Elhamifar (2020b), Considering the cross-dataset bias between ImageNet and ZSL benchmarks Chen et al. (2021a), embedding-based methods were recently proposed to focus on learning the region-based visual features to enhance the holistic visual features[1] using attention mechanism Yu et al. (2018); Zhu et al. (2019); Xie et al. (2019); Huynh & Elhamifar (2020a); Xu et al. (2020). However, these methods depends on the attribute features for attribute localization. As such, Wang et al. (2021) introduces DPPN to iteratively enhance visual features using the category/visual prototype as supervision. Differently, we jointly and mutually refine the visual features and semantic prototype, enabling significant visual-semantic interactions in generative ZSL.

**Generative Zero-Shot Learning.** Since embedding-based methods learn the ZSL classifier only on seen classes, inevitably resulting in the models overfitting seen classes. To tackle this challenge, generative ZSL methods employ the generative models (*e.g.*, VAE and GAN) to generate the unseen visual features for data augmentation Arora et al. (2018); Xian et al. (2018); Shen et al. (2020), ZSL is then converted to a supervised classification task further. As such, the generative ZSL methods have shown significant performance and become very popular recently. However, there are lots of challenges that need to be tackled in generative ZSL. To improve the optimization process of ZSL methods, Skorokhodov & Elhoseiny (2021) introduces class normalization for the ZSL task. Çetin et al. (2022) introduces closed-form sample probing for generative ZSL. To avoid ZSL models overfitting to seen classes, Liu et al. (2021) introduces an isometric propagation network to strengthen the relation between classes within the visual and semantic spaces. Chou et al. (2021) designs a generative scheme to simultaneously generate virtual class labels and their visual features. To enhance the visual feature in generative ZSL methods, Felix et al. (2018), Narayan et al. (2020) and Chen et al. (2021a) refine the holistic visual features by learning a visual→semantic projection, which encourages conditional generator to synthesize visual features with more semantic information. Orthogonal to these methods, we propose a novel dynamic semantic prototype learning method to tackle the visual-semantic domain shift problem and advance generative ZSL further.

**Visual-Semantic Domain Shift vs Projection Domain Shift.** ZSL learns a projection between visual and semantic feature space on seen classes, and then the projection is transferred to the unseen classes for classification. However, data samples of the seen classes (source domain) and unseen classes (target domain) are disjoint, unrelated for some classes, and their distributions can be different, resulting in a large domain shift Fu et al. (2015); Wan et al. (2019); Pourpanah et al. (2022). This challenge is the well-known problem of projection domain shift. To tackle this challenge, inductive-based methods incorporate additional constraints or information from the seen classes Zhang et al. (2019); Jia et al. (2020); Wang et al. (2021). Besides that, several transductive-based methods have been developed to alleviate the projection domain shift problem Fu et al. (2015); Xu et al. (2017); Wan et al. (2019). We should note that the visual-semantic domain shift is different from the projection domain shift. Visual-semantic domain shift denotes that the two domains (visual and semantic domains) share the same categories and only the joint distribution of input and output differs between the training and test stages. In contrast, the projection domain shift can be directly observed in terms of the projection shift, rather than the feature distribution shift Pourpanah et al. (2022). As shown in Fig. 1, we demonstrate that visual-semantic domain shift is a bottleneck challenge in generative ZSL and is neglected by existing methods.

## 3 DYNAMIC SEMANTIC PROTOTYPE LEARNING

**Problem Setting:** The problem setting of ZSL is formulated as follows. Assume that we have seen class data $\mathcal{D}^s = \{(x_i^s, y_i^s)\}$ with $C^s$ seen classes, where $x_i^s \in \mathcal{X}$ denotes the $i$-th visual feature with 2048-dim extracted from a CNN backbone (*e.g.*, ResNet101 He et al. (2016)), and $y_i^s \in \mathcal{Y}^s$ is the corresponding class label. Note that $\mathcal{D}^s$ is divided into training set $\mathcal{D}_{tr}^s$ and test set $\mathcal{D}_{te}^s$ according to Xian et al. (2019a). Another set of unseen classes $C^u$ is $\mathcal{D}_{te}^u = \{(x_i^u, y_i^u)\}$, where $x_i^u \in \mathcal{X}$ are the visual features of unseen classes, and $y_i^u \in \mathcal{Y}^u$ are the corresponding labels. A set of class semantic vectors (semantic values annotated by humans according to attributes) of the class $c \in \mathcal{C}^s \cup \mathcal{C}^u$ with

---

[1]Holistic visual features directly extract from a CNN backbone (*e.g.*, ResNet101 He et al. (2016)) pretrained on ImageNet.

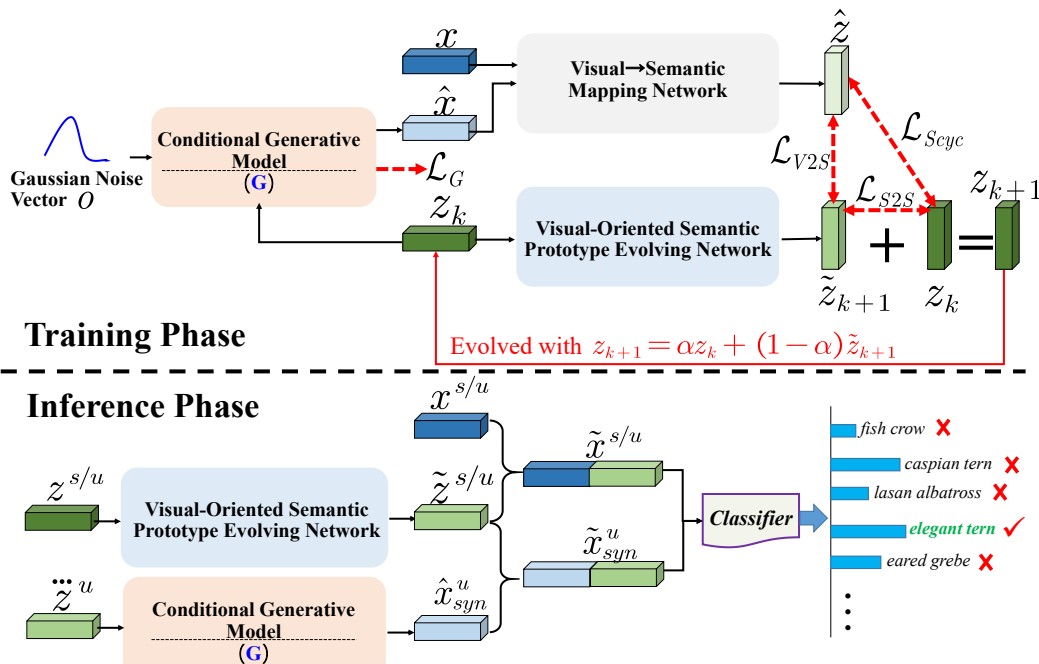

Figure 2: Framework of the dynamic semantic prototype learning (DSP). **Traning Phase** (Top): DSP includes a visual→semantic mapping network (V2SM) and a visual-oriented semantic prototype evolving network (VOPE). V2SM maps visual features into the class semantic space, enabling the conditional generator synthesizes visual samples with rich semantic information and conducts visual information into the VOPE. VOPE iteratively evolves the predefined class semantic prototype under the supervision of visual information. Our DSP is jointly trained with the generative model. **Inference Phase** (Bottom):DSP extracts the evolved semantic prototypes $\tilde{z}^{s/u}$ for seen/unseen classes using the pretrained VOPE, and take the dynamic semantic prototypes $\dddot{z}^u = \alpha z^u + (1-\alpha)\tilde{z}^u$ as input of the conditional generator to synthesize class-specific visual feature samples $\hat{x}^u_{syn}$ for unseen classes. Then, DSP concatenates visual features with their corresponding $\tilde{z}^{s/u}$ to enhance visual feature representations, which are utilized for classification.

$|A|$ attributes, denoted as $z^c = [z^c(1), \ldots, z^c(A)]^\top \in \mathbb{R}^{|A|}$. In the conventional zero-shot learning (CZSL) setting, we aim to learn a classifier for unseen classes, *i.e.*, $f_{CZSL} : X \to Y^U$. In contrast, we aim to learn a classifier for seen and unseen classes in the generalized zero-shot learning (GZSL) setting, *i.e.*, $f_{GZSL} : X \to Y^U \cup Y^S$.

In the following, we first briefly introduce the generative ZSL model. Then, we specifically demonstrate our dynamic semantic prototype learning method, its framework is shown in Fig. 2).

### 3.1 GENERATIVE ZSL MODEL

In our work, we focus on generative ZSL Xian et al. (2018; 2019b). The main goal of generative ZSL is to learn a conditional generator G: $\mathcal{Z} \times \mathcal{O} \to \tilde{\mathcal{X}}$, which takes the class semantic vectors/prototypes $z \in \mathcal{Z}$ and Gaussian noise vectors $o \in \mathcal{O}$ as inputs, and synthesizes the class-specific visual feature samples $\tilde{x} \in \tilde{\mathcal{X}}$. Once such a conditional generative model is learned, we can take it to synthesizes a large number of class-specific visual feature samples for unseen classes based on the semantic vector $z^u$, and the final classifier over $y \in \mathcal{Y}$ can be obtained using any standard supervised classifier (*e.g.*, softmax).

### 3.2 VISUAL→SEMANTIC MAPPING NETWORK

To encourage the conditional generator (G) to synthesize class-specific visual samples with enriched semantic information and conduct visual information for evolving the semantic prototype, we design a Visual→Semantic Mapping Network (V2SM) that is entailed on the G of the conditional generative model. V2SM is a multilayer perceptron (MLP), which has a residual block to avoid too much

information loss when mapping visual features of 2048-dim to semantic features of $|A|$-dim. The network details are presented in Appendix A. Specifically, V2SM maps the real/synthesized visual features $x/\hat{x}$ as semantic vectors $\hat{z}_{real}/\hat{z}_{syn}$, respectively, formulated as:

$$\hat{z}_{real} = \text{V2SM}(x) \quad \text{or} \quad \hat{z}_{syn} = \text{V2SM}(\hat{x}), \tag{1}$$

where $\hat{z}_{real} \cup \hat{z}_{syn} = \hat{z}$, and $\hat{z} \in \mathbb{R}^{|A|}$. We can find that the conditional generator (*i.e.*, semantic→visual mapping) and V2SM conduct a semantic cycle network, *i.e.*, $z \to G(o, z) = \hat{x}$ and $\hat{x} \to \text{V2SM}(\hat{x}) = \hat{z}$, where $o$ is a random Gaussian noise vector with $|A|$-dim. As such, V2SM encourages the conditional generator to synthesize visual features with more semantic information. To enable V2SM to map visual features into semantic space effectively, we employ the semantic cycle-consistency loss ($\mathcal{L}_{Scyc}$) during training. This loss can be written as:

$$\mathcal{L}_{Scyc} = \mathbb{E}\left[\|\hat{z}_{real} - z_k\|_1\right] + \mathbb{E}\left[\|\hat{z}_{syn} - z_k\|_1\right], \tag{2}$$

where $z_k$ is the dynamic semantic prototype learned by VOPE at the $k$-th step, which will be introduced in the next subsection. This is different from TF-VAEGAN Narayan et al. (2020) and FREE Chen et al. (2021a) that use the predefined semantic prototypes as semantic supervision in their semantic decoders. Since the dynamic semantic prototype is more close to its corresponding visual prototype, it provides more accurate supervision signals for the V2SM and generative models. As such, DSP can significantly improve the performance of TF-VAEGAN and FREE .

We find that $z$, $\hat{z}$ and $z_k$ are closely affected by each other. $\hat{z} = \text{V2SM}(x)$ is mapped from the visual features in V2SM and used to convey visual signal to supervise VOPE, enabling $z$ to be progressively evolved as dynamic semantic prototype $z_k = \alpha z + (1-\alpha)\text{VOPE}(z_{k-1})$ ($z_k = z$ when k=0). As such, $z_k$ IS closer to its corresponding visual prototype, and serves as the accurate condition information that encourages generator to synthesize reliable visual features.

## 3.3 VISUAL-ORIENTED SEMANTIC PROTOTYPE EVOLVING NETWORK

We further introduce a visual-oriented semantic prototype evolving network (VOPE) to refine the semantic prototype under the supervision of visual information. As such, the dynamic semantic prototype can be closer to its corresponding visual prototype and the visual-semantic domain shift can be alleviated, as shown in Fig. 1(c). Considering that the predefined semantic prototypes have rich accurate information annotated by humans, we progressively update it progressively based on the visual information guidance. Targeting this goal, VOPE is designed as an MLP network with a residual block that acts as a routing gate mechanism implemented by channel attention, which evolves the specific attribute value of semantic prototypes according to the corresponding visual information. The network details are presented in Appendix B.

Specifically, VOPE iteratively evolves the initialized semantic prototype $z_k$ ($z_k$ is the predefined semantic prototype $z$ annotated by humans when k=0) to another semantic prototype space $\tilde{z}$, formulated as:

$$\tilde{z}_{k+1} = \text{VOPE}(z_k) \tag{3}$$

To encourages $\tilde{z}_{k+1}$ evolved with visual supervision information, we conduct the visual signals from V2SM, *e.g.*, $\hat{z}$. We formulate $\hat{z}$ as supervision using a visual-to-semantic loss:

$$\mathcal{L}_{V2S} = \mathbb{E}\left[1 - cosine(\hat{z}, \tilde{z}_{k+1})\right]. \tag{4}$$

Here, we use cosine similarity as a weak constraint for VOPE to conduct the information transfer from the visual domain to the semantic domain. The reason is that we do not desire the visual information to be over strong, helping the semantic prototype evolve progressively. Simultaneously, we also use the semantic reconstruction loss to constrain VOPE, defined as:

$$\mathcal{L}_{S2S} = \mathbb{E}\left[\|\tilde{z}_{k+1} - z_k\|_1\right], \tag{5}$$

where $z_{k+1}$ is the dynamic semantic prototype at the $k + 1$-th step, formulated as follows:

$$z_{k+1} = \alpha z_k + (1-\alpha)\tilde{z}_{k+1}, \tag{6}$$

where $\alpha$ is a combination coefficient for fusing the dynamic semantic prototype $z_k$ evolved in the previous step and the mapped semantic vectors $\tilde{z}_{k+1}$ of the current step in VOPE. Thus, $z_k$ is dynamically evolved for each training step and fed back to VOPE as input. Considering that the empirically predefined semantic prototype provides rich semantic priori, $\alpha$ is set to relatively large (*i.e.*, $\alpha = 0.9$) for preserving its original information. Such a setting i) encourages the dynamic semantic prototype to evolve progressively under the supervision of visual information, and ii) facilitates model optimization .

### 3.4 OPTIMIZATION

Our DSP is jointly optimized with the generative model, e.g., CLSWGAN Xian et al. (2018), f-VAEGAN Xian et al. (2019b), TF-VAEGAN Narayan et al. (2020) and FREE Chen et al. (2021a), with the following objective function:

$$\mathcal{L}_{total} = \mathcal{L}_G + \lambda_{Scyc}\mathcal{L}_{Scyc} + \lambda_{V2S}\mathcal{L}_{V2S} + \lambda_{S2S}\mathcal{L}_{S2S}, \tag{7}$$

where $\mathcal{L}_G$ is the loss of generative model, $\lambda_{Scyc}$, $\lambda_{V2S}$, $\lambda_{S2S}$ are the weights that control the importance of the related loss terms. Since $\lambda_{Scyc}$ and $\lambda_{S2S}$ are semantic reconstruction losses, we set $\lambda_{Scyc} = \lambda_{S2S}$. With the similar measurement loss terms and model components designed for DSP, we empirically observe that our model is robust and easy to train on various ZSL benchmarks when it is entailed on various generative models.

### 3.5 INFERENCE PHASE

**Synthesizing Visual Features for Unseen Classes:** After training, we first take the pre-trained VOPE to update the predefined semantic prototypes to be dynamic semantic prototypes for unseen classes using Eq. 3 and Eq. 6, *i.e.*, $\ddot{z}^u = \alpha z^u + (1 - \alpha)\text{VOPE}(z^u)$. Then, we take the dynamic semantic prototypes of unseen classes as conditional supervision in generator (G) to synthesize class-specific visual features for unseen classes:

$$\hat{x}^u_{syn} = G(o, \ddot{z}^u), \tag{8}$$

where $o$ is a random Gaussian noise vector with $|A|$-dim.

**Enhancing Visual Features for All Classes:** Finally, we concatenate visual features (including the real visual features of seen classes and real/synthesized visual features of unseen classes) and their corresponding evolved semantic prototypes $\tilde{z} = \text{VOPE}(z)$ to enhance them, enabling the visual features to be closer to their corresponding semantic prototypes and alleviate the cross-dataset bias Chen et al. (2021a). It is formulated as:

$$\textcolor{red}{\textbf{Seen Classes}} : \tilde{x}^s_{tr} = x^s_{tr} \oplus \tilde{z}^s; \qquad \tilde{x}^s_{te} = x^s_{te} \oplus \tilde{z}^s, \tag{9}$$

$$\textcolor{red}{\textbf{Unseen Classes}} : \tilde{x}^u_{syn} = \hat{x}^u_{syn} \oplus \tilde{z}^u; \quad \tilde{x}^u_{te} = x^u_{te} \oplus \tilde{z}^u, \tag{10}$$

where $\tilde{z}^s \cup \tilde{z}^u = \tilde{z}$, $x_{tr}$ is the real visual feature in the training set $\mathcal{D}^s_{tr}$, $x_{te}$ is the real visual feature in the test sets $\mathcal{D}^s_{te}$ and $\mathcal{D}^u_{te}$. As such, the enhanced visual features are closer to their corresponding semantic prototypes, and they are more discriminative as the cross-dataset bias is alleviated. These enhanced visual features are used for classification later.

**ZSL Classification:** We employ $\tilde{x}^s_{tr}$ and $\tilde{x}^u_{syn}$ to lean a classifier (*e.g.*, softmax), *i.e.*, $f_{czsl} : \tilde{\mathcal{X}} \rightarrow \mathcal{Y}^s \cup \mathcal{Y}^u$ or $f_{gzsl} : \tilde{\mathcal{X}} \rightarrow \mathcal{Y}^s \cup \mathcal{Y}^u$. Once the classifier is trained, we use $\tilde{x}^s_{te}$ and $\tilde{x}^u_{te}$ to test the model further. Note that our DSP is an inductive method as we do not use the real visual features of unseen classes for model optimization.

### 4 EXPERIMENTS

**Datasets.** We evaluate the effectiveness of our DSP on three well-known ZSL benchmark datasets, *i.e.*, two fine-grained datasets ( CUB Welinder et al. (2010), SUN Patterson & Hays (2012) and FLO Nilsback & Zisserman (2008)) and one coarse-grained dataset (AWA2 Xian et al. (2019a)). CUB consists of 11,788 images of 200 bird classes (seen/unseen classes = 150/50) captured by 312 attributes. SUN includes 14,340 images of 717 scene classes (seen/unseen classes = 645/72) described by 102 attributes. AWA2 contains 37,322 images of 50 animal classes (seen/unseen classes = 40/10) captured by 85 attributes. FLO inludes 8,189 images of 102 flower classes (seen/unseen classes = 82/20) captured by 1024 attributes.

**Implementation Details.** We use the training splits proposed in Xian et al. (2019a). Meanwhile, the visual features of images (input size $224 \times 224$) are extracted from the 2048-dimensional top-layer pooling units of a CNN backbone (i.e., ResNet-101 He et al. (2016)) pre-trained on ImageNet. The V2SM and VOPE are MLPs with residual blocks, network details of which can be found in Appendix A and Appendix B, respectively. Since f-VAEGAN Xian et al. (2019b) is a popular generative ZSL method, we take it as a baseline and entail our DSP on it for conducting the ablation study, qualitative experiment and hyper-parameter analysis. Following Xian et al. (2019a), we also use the same validation split on each dataset to conduct the hyperparameter setting. We synthesize 150, 800

Table 1: Compared our DSP with the state-of-the-arts on CUB, SUN, AWA2 and FLO benchmark datasets in the GZSL setting. ♣ denotes embedding-based methods, while ♠ denotes generative methods. The best and second-best results are marked in **Red** and **Blue**, respectively.

| | Methods | CUB | | | SUN | | | AWA2 | | | FLO | | |
|---|---|---|---|---|---|---|---|---|---|---|---|---|---|
| | | U | S | H | U | S | H | U | S | H | U | S | H |
| ♣ | SGMA Zhu et al. (2019) | 36.7 | 71.3 | 48.5 | – | – | – | 37.6 | 87.1 | 52.5 | – | – | – |
| | AREN Xie et al. (2019) | 38.9 | **78.7** | 52.1 | 19.0 | 38.8 | 25.5 | 15.6 | **92.9** | 26.7 | – | – | – |
| | APN Xu et al. (2020) | **65.3** | 69.3 | **67.2** | 41.9 | 34.0 | 37.6 | 56.5 | 78.0 | 65.5 | – | – | – |
| | DAZLE Huynh & Elhamifar (2020a) | 56.7 | 59.6 | 58.1 | **52.3** | 24.3 | 33.2 | 60.3 | 75.7 | 67.1 | – | – | – |
| | CN Skorokhodov & Elhoseiny (2021) | 49.9 | 50.7 | 50.3 | 44.7 | **41.6** | 43.1 | 60.2 | 77.1 | 67.6 | – | – | – |
| | I2DFormer Naeem et al. (2022) | 35.3 | 57.6 | 43.8 | – | – | – | **66.8** | 76.8 | **71.5** | 35.8 | **91.9** | 51.5 |
| ♠ | f-VAEGAN Xian et al. (2019b) | 48.7 | 58.0 | 52.9 | 45.1 | 38.0 | 41.3 | 57.6 | 70.6 | 63.5 | 56.8 | 74.9 | 64.6 |
| | TF-VAEGAN Narayan et al. (2020) | 53.7 | 61.9 | 57.5 | 48.5 | 37.2 | 42.1 | 58.7 | 76.1 | 66.3 | 62.5 | 84.1 | 71.7 |
| | LsrGAN Vyas et al. (2020) | 48.1 | 59.1 | 53.0 | 44.8 | 37.7 | 40.9 | 54.6 | 74.6 | 63.0 | – | – | – |
| | AGZSL Chou et al. (2021) | 48.3 | 58.9 | 53.1 | 29.9 | 40.2 | 34.3 | **65.1** | 78.9 | 71.3 | – | – | – |
| | FREE Chen et al. (2021a) | 54.9 | 60.8 | 57.7 | 47.4 | 37.2 | 41.7 | 60.4 | 75.4 | 67.1 | **67.4** | 84.5 | **75.0** |
| | GCM-CF Yue et al. (2021) | 61.0 | 59.7 | 60.3 | 47.9 | 37.8 | 42.2 | 60.4 | 75.1 | 67.0 | – | – | – |
| | HSVA Chen et al. (2021b) | 52.7 | 58.3 | 55.3 | 48.6 | 39.0 | **43.3** | 59.3 | 76.6 | 66.8 | – | – | – |
| | FREE+ESZSL Çetin et al. (2022) | 51.6 | 60.4 | 55.7 | 48.2 | 36.5 | 41.5 | 51.3 | 78.0 | 61.8 | 65.6 | 82.2 | 72.9 |
| | TF-VAEGAN+ESZSL Çetin et al. (2022) | 51.1 | 63.3 | 56.6 | 44.0 | 39.7 | 41.7 | 55.2 | 74.7 | 63.5 | 63.5 | 83.2 | 72.1 |
| | f-VAEGAN Xian et al. (2019b)+**DSP** | **62.5** | **73.1** | **67.4** | **57.7** | 41.3 | **48.1** | 63.7 | **88.8** | **74.2** | 66.2 | 86.9 | **75.2** |

and 3400 visual features per unseen class to train the classifier for SUN, CUB and AWA2 datasets, respectively. The combination coefficient for updating semantic prototypes is set to 0.9 for all datasets. The loss weight of $\lambda_{Scyc} = \lambda_{S2S}$ is set to 0.1, 0.01 and 0.001 for CUB, SUN and AWA2, respectively. The loss weight of $\lambda_{V2S}$ is set to 0.6/0.6 and 1.0 for CUB/AWA2 and SUN, respectively. The specific hyper-parameter settings of our DSP entailed on other baselines are in Appendix C.

**Evaluation Protocols.** During testing (ZSL classification), we follow the unified evaluation protocols proposed in Xian et al. (2019a). We calculate the top-1 accuracy of the unseen class for the CZSL setting (denoted as $Acc$). In the GZSL setting, we measure the top-1 accuracy on seen and unseen classes, denoted as $S$ and $U$, respectively. Their harmonic mean (defined as $H = (2 \times S \times U)/(S + U)$) are a better protocols in the GZSL setting. As such, we typically take the $H$ to discuss the performance of ZSL methods in the GZSL setting.

## 4.1 EXPERIMENTAL RESULTS

**Comparing with the SOTA.** We first compare our f-VAEGAN+DSP with the state-of-the-art methods in the GZSL setting. Table 1 shows the performance of various ZSL methods, including non-generative methods (*e.g.*, SGMA Zhu et al. (2019), AREN Xie et al. (2019), APN Xu et al. (2020), DAZLE Huynh & Elhamifar (2020a), CN Skorokhodov & Elhoseiny (2021), I2DFormer Naeem et al. (2022)) and generative methods (*e.g.*, f-VAEGAN Xian et al. (2019b), TF-VAEGAN Narayan et al. (2020), AGZSL Chou et al. (2021), GCM-CF Chou et al. (2021), HSVA Chen et al. (2021b), ESZSL Çetin et al. (2022)). Compared to the popular non-generative methods, our f-VAEGAN+DSP achieves the best results on all datasets, *i.e.*, CUB ($H = 67.4$), SUN ($H = 48.1$), AWA2 ($H = 74.2$) and FLO ($H = 75.2$). Thus, our DSP effectively helps generative ZSL methods beat the embedding-based methods. Our method also achieves the best results with significant improvements compared to the other generative methods. For instance, our f-VAEGAN+DSP outperforms the latest generative method (*i.e.*, TF-VAEGAN+ESZSL Çetin et al. (2022)) by a large-margin, resulting in the improvements of harmonic mean by 10.8%, 6.4% and 10.7% on CUB, SUN and AWA2, respectively. These results consistently demonstrate the superiority and great potential of our DSP in generative ZSL.

Furthermore, we also compare our DSP with the popular generative methods in the CZSL setting, as shown in Table 2. f-VAEGAN+DSP achieves competitive performances of harmonic mean with 62.8, 68.6 and 71.6 on CUB, SUN and AWA2, respectively. These results indicate that DSP also effectively tackles the semantic-visual domain shift problem of the generative ZSL in the CZSL setting.

Table 2: Compared our method with the generative ZSL methods in the CZSL setting.

| Methods | CUB | SUN | AWA2 |
|---|---|---|---|
| | acc | acc | acc |
| CLSWGAN Xian et al. (2018) | 57.3 | 60.8 | 68.2 |
| f-VAEGAN Xian et al. (2019b) | 61.0 | 64.7 | 71.1 |
| CADA-VAE Schönfeld et al. (2019) | 59.8 | 61.7 | 63.0 |
| Composer Narayan et al. (2020) | **69.4** | 62.6 | 71.5 |
| FREE Chen et al. (2021a) | 64.8 | 65.0 | 68.9 |
| HSVA Chen et al. (2021b) | 62.8 | 63.8 | 70.6 |
| f-VAEGAN + **DSP** | 62.8 | **68.6** | **71.6** |

**Ablation Study.** Based on the baseline (*i.e.*, f-VAEGAN Xian et al. (2019b)), we conduct ablation studies to evaluate the effectiveness of our DPS in terms of the $\mathcal{L}_{Scyc}$, $\mathcal{L}_{V2S}$, $\mathcal{L}_{S2S}$, semantic prototype evolution (*i.e.*, Eq. 6) and enhancing visual feature (*i.e.*, Eq. 9 and Eq. 10). Our results are shown in Table 3. When DSP is without the constraints of $\mathcal{L}_{Scyc}$, its performance is dropped

Table 3: Ablation studies for different components of DSP. The *baseline* is the popular generative ZSL model, *i.e.*, f-VAEGAN Xian et al. (2019b).

| Method | CUB | | | SUN | | |
|---|---|---|---|---|---|---|
| | U | S | H | U | S | H |
| baseline | 48.7 | 58.0 | 52.9 | 45.1 | 38.0 | 41.3 |
| baseline+DSP (w/o $\mathcal{L}_{Scyc}$) | 60.0 | 63.9 | 61.9 | 57.4 | 38.4 | 46.0 |
| baseline+DSP (w/o $\mathcal{L}_{S2S}$) | 61.9 | 69.6 | 65.5 | 58.1 | 37.2 | 45.4 |
| baseline+DSP (w/o $\mathcal{L}_{V2S}$) | 58.8 | 61.0 | 59.9 | 55.1 | 34.8 | 42.7 |
| baseline+DSP (w/o Eq. 6) | 54.4 | 55.3 | 54.8 | 50.3 | 36.2 | 42.1 |
| baseline+DSP (w/o Eq. 9 and Eq. 10) | 52.4 | 53.9 | 53.1 | 54.2 | 35.0 | 42.5 |
| baseline+DSP (full) | 62.5 | 73.1 | 67.4 | 57.7 | 41.3 | 48.1 |

as the generator in the generative model lacks semantic cycle-consistency supervision. Meanwhile, the performance of DSP is dropped slightly without the optimization of $\mathcal{L}_{S2S}$. In contrast, if DSP can not be evolved under the supervision of visual information using $\mathcal{L}_{V2S}$, DSP achieves very poor results compared to its full model, *i.e.*, the harmonic mean drops by 7.5% and 5.4% on CUB and SUN, respectively. These results demonstrate that evolving semantic prototypes according to the visual information is effective for refining the predefined semantic prototypes aligned to the visual prototypes, helping DSP tackle the visual-semantic domain shift problem in generative ZSL. Notably, if DSP does not dynamically evolve the predefined semantic prototypes for generative learning, *i.e.*, DSP w/o Eq. 6, it almost loses its effectiveness and only achieves little improvement for the baseline with additional constraints (*i.e.*, $\mathcal{L}_{Scyc}$, $\mathcal{L}_{V2S}$ and $\mathcal{L}_{S2S}$). For instance, its performance of harmonic mean drops by 12.6% and 6.0% on CUB and SUN, respectively, compared to its full model. Enhancing visual feature also effectively improves the performance of DSP. These results show that the dynamic semantic prototype is effective to tackle the visual-semantic domain shift.

Table 4: Evaluation of DSP with multiple popular generative ZSL models on three benchmark datasets. Each row pair shows the effect of adding DSP to a particular generative ZSL model.

| Methods | CUB | | | SUN | | | AWA2 | | |
|---|---|---|---|---|---|---|---|---|---|
| | U | S | H | U | S | H | U | S | H |
| CLSWGAN Xian et al. (2018) | 43.7 | 57.7 | 49.7 | 42.6 | 36.6 | 39.4 | 57.9 | 61.4 | 59.6 |
| CLSWGAN Xian et al. (2018)+**DSP** | **51.4** | **63.8** | **56.9**$^{\uparrow 7.2}$ | **48.3** | **43.0** | **45.5**$^{\uparrow 6.1}$ | **60.0** | **86.0** | **70.7**$^{\uparrow 11.1}$ |
| f-VAEGAN Xian et al. (2019b) | 48.7 | 58.0 | 52.9 | 45.1 | 38.0 | 41.3 | 57.6 | 70.6 | 63.5 |
| f-VAEGAN Xian et al. (2019b)+**DSP** | **62.5** | **73.1** | **67.4**$^{\uparrow 14.5}$ | **57.7** | **41.3** | **48.1**$^{\uparrow 6.8}$ | **63.7** | **88.8** | **74.2**$^{\uparrow 10.7}$ |
| TF-VAEGAN Narayan et al. (2020) | 53.7 | 61.9 | 57.5 | 48.5 | 37.2 | 42.1 | 58.7 | 76.1 | 66.3 |
| TF-VAEGAN Narayan et al. (2020)+**DSP** | **58.7** | **67.4** | **62.8**$^{\uparrow 5.3}$ | **60.3** | **45.3** | **51.7**$^{\uparrow 9.6}$ | **65.6** | **87.1** | **74.8**$^{\uparrow 8.5}$ |
| FREE Chen et al. (2021a) | 54.9 | 60.8 | 57.7 | 47.4 | 37.2 | 41.7 | 60.4 | 75.4 | 67.1 |
| FREE Chen et al. (2021a)+**DSP** | **60.9** | **68.7** | **64.6**$^{\uparrow 6.9}$ | **60.3** | **44.1** | **51.0**$^{\uparrow 9.3}$ | **65.3** | **89.2** | **75.4**$^{\uparrow 8.3}$ |

**Generative ZSL Methods with DSP.** To evaluate our dynamic semantic prototype learning method as a general technique to improve generative model, we entail it into four recent popular generative ZSL methods, *i.e.*, CLSWGAN Xian et al. (2018), f-VAEGAN Xian et al. (2019b), TF-VAEGAN Narayan et al. (2020) and FREE Chen et al. (2021a). We use the official repositories shared by respective authors of these works to implement experimental results. Since TF-VAEGAN and FREE have the semantic decoder, we only add the VOPE with $\mathcal{L}_{V2S}$ and $S2S$ to enable them to learn dynamic semantic prototypes. The specific hyper-parameter settings of our DSP entailed on these baselines are in Appendix C. As shown in Table 4, DSP consistently improves the performances of all baselines on all benchmark datasets by a large-margin. For example, the average performance gains of harmonic mean are 8.5%, 8.0% and 9.7% on CUB, SUN and AWA2, respectively. These results demonstrate that *i)* the visual-semantic domain shift is an important bottleneck challenge and neglected by existing generative ZSL, and ii) our dynamic semantic prototype learning method effectively tackles this problem.

**Qualitative Performance.** To intuitively show the effectiveness of our DSP, we qualitatively investigate the improvements over baseline (*i.e.*, f-VAEGAN Xian et al. (2019b)). We conduct t-SNE visualization Maaten & Hinton (2008) of the real/synthesized visual features learned by the f-VAEGAN and f-VAEGAN+DSP on 10 classes randomly selected from CUB. As shown in Fig. 3(Left), the visual features synthesized by f-VAEGAN Xian et al. (2019b) are far away from their corresponding real visual features, caused by the visual-semantic domain shift problem. Meanwhile, since the cross-dataset bias problem Chen et al. (2021a), the real visual features directly extracted

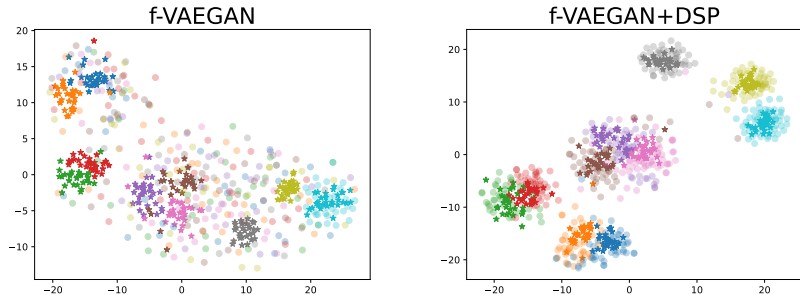

Figure 3: t-SNE visualizations of visual features learned by the f-VAEGAN Xian et al. (2019b) (Left) and f-VAEGAN+DSP (Right). The 10 colors denote 10 different classes randomly selected from CUB. The "○" and "⋆" indicate the real and synthesized visual features, respectively. The t-SNE visualizations on SUN and AWA2 are in Appendix D. (Best viewed in color)

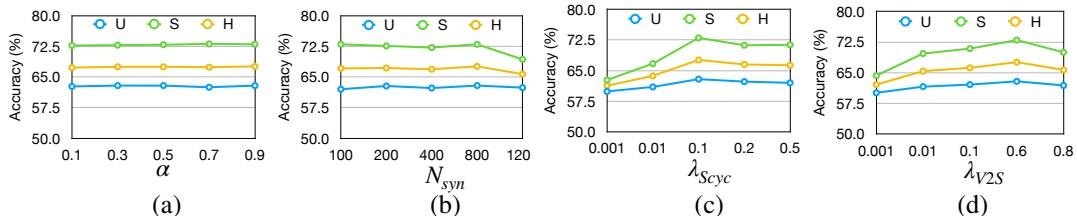

Figure 4: The effects of (a) combination coefficient $\alpha$, (b) synthesizing number of per unseen classes $N_{syn}$, (c) loss weights $\lambda_{Scyc}$ and (d) loss weights $\lambda_{V2S}$. We take CUB as an example.

from the CNN backbone are poor quality. In contrast, our DSP dynamically evolves the predefined semantic prototypes to accurately represent the visual representations, enabling the generator to synthesize reliable visual features for all classes (as shown in Fig. 3(Right)). Furthermore, DSP enhances the real/synthesize visual features by enriching semantic information into visual representations. As such, DSP obtains significant performance gains over f-VAEGAN Xian et al. (2019b), *i.e.*, considerable improvements of harmonic mean with 14.5%, 6.8% and 10.7% on CUB, SUN and AWA2, respectively.

**Hyper-Parameter Analysis.** We conduct extensive experiments to investigate the effects of various hyper-parameter on CUB, *i.e.*, combination coefficient $\alpha$ in Eq. 6, synthesizing number of per unseen classes $N_{syn}$, the loss weights $\lambda_{Scyc}$ and $\lambda_{V2S}$. Since $\mathcal{L}_{Scyc}$ and $\mathcal{L}_{S2S}$ are the semantic reconstruction loss, we set $\lambda_{Scyc} = \lambda_{S2S}$ and take one loss weight into the analysis. From the results in Fig. 4(a), DSP is not insensitive to the combination coefficient for dynamic semantic prototypes evolution, and DSP achieves better results with a relatively large value of $\alpha$ (*i.e.*, 0.9). The reason is that the dynamic semantic prototypes can adapt to the evolution according to the visual-semantic interactions during optimization. The results shown in Fig. 4(b) indicate our DSP is robust to the $N_{syn}$ when $N_{syn}$ is not set too large. Results in Fig. 4(c) and Fig. 4(d) show the best effects of our f-VAEGAN+DSP can be achieved when $\lambda_{Scyc}$ and $\lambda_{V2S}$ is set to 0.1 and 0.6, respectively. Additionally, we find that $\lambda_{V2S}$ should be set to larger than $\lambda_{Scyc}$ and $\lambda_{S2S}$. This is because the dynamic semantic prototypes should mainly depend on the visual information during evolution. Overall, our model is robust to all hyper-parameters and easy to optimize.

## 5 CONCLUSION

In this work, we are the first to discuss that the predefined semantic prototype results in the visual-semantic domain shift problem in generative ZSL. To tackle this problem and advance generative ZSL, we propose a dynamic semantic prototype learning method, which jointly and mutually refines the semantic prototypes and visual features enabling the generator to synthesize reliable visual features. Our DSP is plug-and-play to be entailed on various baselines and consistently achieves large-margin improvements on three ZSL benchmark datasets. We believe that our work will facilitate the development of stronger zero-shot systems and motivate their deployment in real-world applications.

**Reproducibility statement.** We discuss the problem of visual-semantic domain shift in generative ZSL and propose dynamic semantic prototype learning to tackle it. We intuitively demonstrate that it is an important bottleneck problem in generative ZSL. We also clearly introduce our model details, implementation details and hyper-parameter settings, which we use in our experiments. We will provide our source code on a public repository once our paper is accepted.

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

**APPENDIX**
Appendix organization:

## A    NEWORK DETAILS OF V2SM

As shown in Fig. 5, the network of V2SM is an MLP with a residual block, which avoids too much information loss when mapping the visual feature of 2048-dim into semantic space with $|A|$-dim.

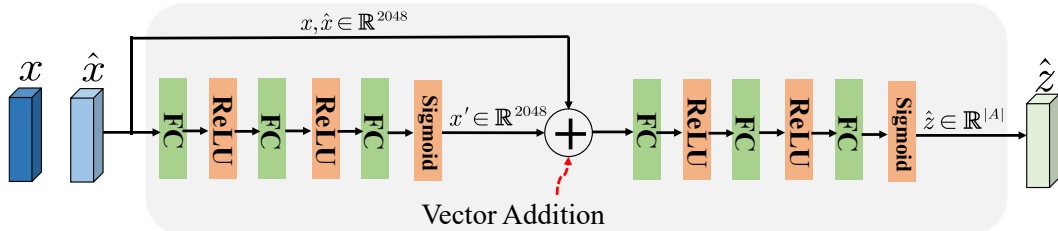

Figure 5: Network details of V2SM.

## B    NEWORK DETAILS OF VOPE

As shown in Fig. 6, the network of VOPE is an MLP with a residual block (fusing with Hardamad Product). Essentially, the network of VOPE is a routing gate mechanism implemented by channel attention, which is effective in selectively evolving the specific attribute value of semantic prototypes according to the corresponding visual information. As such, the predefined semantic prototypes can be progressively evolved and aligned to the visual prototypes.

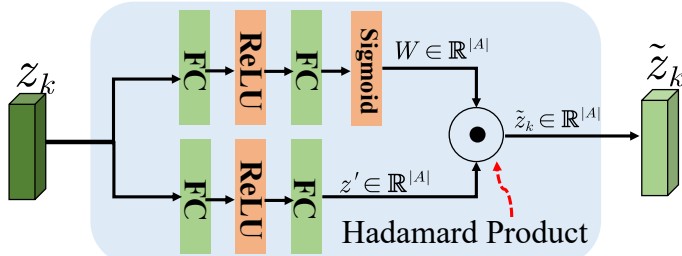

Figure 6: Network details of VOPE.

## C    THE HYPER-PARAMETER SETTINGS OF OUR DSP ENTAILED ON VARIOUS BASELINES

We present the hyper-parameter settings of our DSP entailed on various baselines (*i.e.*, CLSWGAN Xian et al. (2018), f-VAEGAN Xian et al. (2019b), TF-VAEGAN Narayan et al. (2020) and FREE Chen et al. (2021a)) on CUB, SUN and AWA2. Including the synthesizing number of per unseen classes $N_{syn}$, the loss weights $\lambda_{Scyc}$, $\lambda_{V2S}$ and combination coefficient $\alpha$ in Eq. 6. We empirically observe that our DSP is robust and easy to train when it is entailed on various generative models. Based on these hyper-parameter settings, our DSP achieves significant performance gains

Table 5: The hyper-parameter settings of our DSP entailed on various baselines (*i.e.*, CLSWGAN Xian et al. (2018), f-VAEGAN Xian et al. (2019b), TF-VAEGAN Narayan et al. (2020) and FREE Chen et al. (2021a)) on CUB, SUN and AWA2. Including the synthesizing number of per unseen classes $N_{syn}$, the loss weights $\lambda_{Scyc}$, $\lambda_{V2S}$ and combination coefficient $\alpha$ in Eq. 6.

| Methods | CUB | | | | SUN | | | | AWA2 | | | |
|---|---|---|---|---|---|---|---|---|---|---|---|---|
| | $N_{syn}$ | $\lambda_{Scyc}$ | $\lambda_{V2S}$ | $\alpha$ | $N_{syn}$ | $\lambda_{Scyc}$ | $\lambda_{V2S}$ | $\alpha$ | $N_{syn}$ | $\lambda_{Scyc}$ | $\lambda_{V2S}$ | $\alpha$ |
| clswGAN Xian et al. (2018) + **DSP** | 300 | 0.15 | 1.0 | 0.9 | 300 | 0.005 | 1.0 | 0.9 | 3400 | 0.1 | 1.0 | 0.9 |
| f-VAEGAN Xian et al. (2019b)+**DSP** | 800 | 0.1 | 0.6 | 0.9 | 150 | 0.01 | 1.0 | 0.9 | 3400 | 0.001 | 0.6 | 0.9 |
| TF-VAEGAN Narayan et al. (2020) + **DSP** | 400 | 0.01 | 1.0 | 0.9 | 500 | 0.05 | 1.5 | 0.9 | 5300 | 0.09 | 1.4 | 0.9 |
| FREE Chen et al. (2021a) + **DSP** | 600 | 0.1 | 0.6 | 0.9 | 150 | 0.01 | 1.0 | 0.9 | 4000 | 0.001 | 2.0 | 0.9 |

over the various popular generative models on all datasets. For instance, the average performance gains of harmonic mean are 8.5%, 8.0% and 9.7% on CUB, SUN and AWA2, respectively. Please refer to Table 4.

# D   T-SNE VISUALIZATION ON SUN AND AWA2

As shown in Fig. 7, t-SNE visualizations of visual features learned by the f-VAEGAN Xian et al. (2019b) (Left) and f-VAEGAN+DSP (Right) on SUN and AWA2. Analogously, the visual features generated by f-VAEGAN are also far away from their corresponding real ones, and the discrimination of these real/synthesized visual features is undesirable. In contrast, our DSP help f-VAEGAN synthesize visual features close to their corresponding real ones and enhance these real visual features. As such, our DSP significantly improves the performances of f-VAEGAN Xian et al. (2019b) on SUN and AWA2.

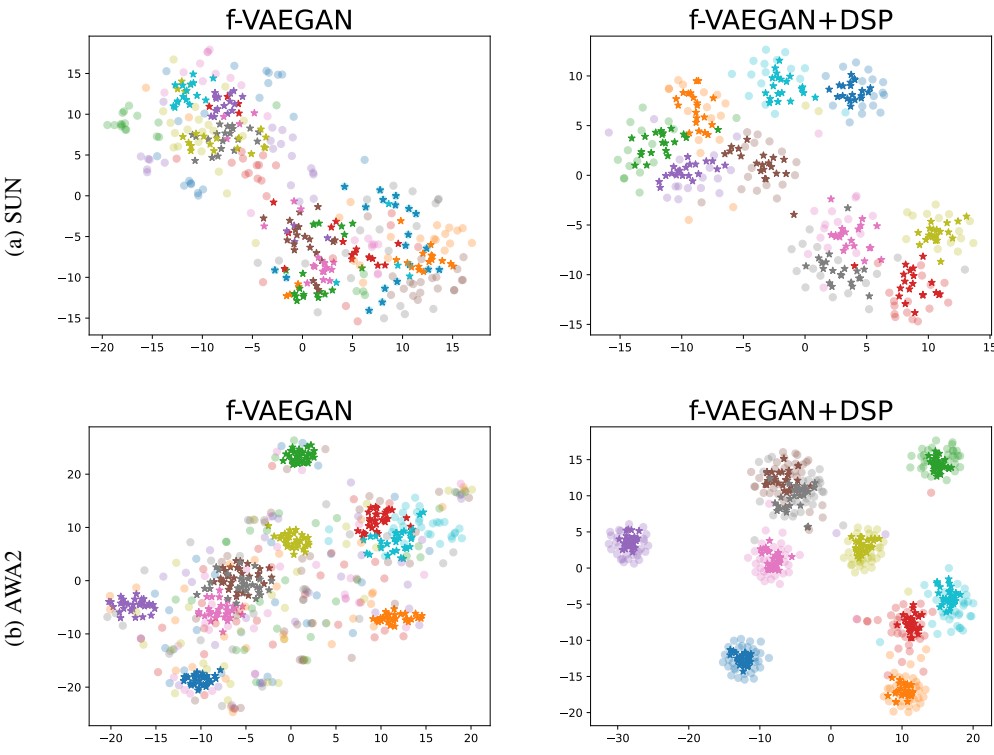

Figure 7: t-SNE visualizations of visual features learned by the f-VAEGAN Xian et al. (2019b) (Left) and f-VAEGAN+DSP (Right) on (a) SUN and (b) AWA2. The "○" and "⋆" indicate the real and synthesized visual features, respectively. (Best viewed in color)

