# OpenReview forum: "DSP: Dynamic Semantic Prototype for Generative Zero-Shot Learning"
_ICLR.cc/2023/Conference — Submitted to ICLR 2023_

### Official Review · Reviewer_J6h5 · 2022-10-22

**Confidence:** 5
**Clarity, Quality, Novelty And Reproducibility:** 1. The proposed dynamic prototypes ar…
**Correctness:** 2
**Technical Novelty And Significance:** 2
**Empirical Novelty And Significance:** 2
**Recommendation:** 5

**Strength And Weaknesses:**

Strength:
1. This paper is well-organized and well-written, which is easy to read.
2. The idea of dynamic prototypes for zero-shot learning is interesting and the prototype evolving is novel.
3. The proposed approach achieves good performance on different baselines.

Weakness:
1. The V2SM network contains six fully connected layers, which introduce many parameters. How does the training efficient? Will it be easy to overfit?
2. What is the motivation to design the structure of VOPE? It is a structure of self-attention, which is not influenced by visual samples.
3. It is unclear how the dynamic prototypes are obtained in the inference phase. The authors said ‘ take the pretrained VOPE to extract the dynamic semantic prototypes for all classes ’, which is strange. In the training stage, the prototypes for the seen classes are evolving with changing parameters. However, in the test stage, they are obtained with fixed parameters, which are not the same. Moreover, only the k+1th prototype is used, is it reasonable to be called a dynamic prototype?
4. In the classification process, the authors concatenate the visual features and the semantic features to augment the features. Why concatenate the features? The influence is not shown.
5. Figure 3 is strange. Why the real samples are scattered in the first picture while they are clustered in the second picture?

**Summary Of The Paper:**

This paper proposes a dynamic semantic prototype learning method, which jointly refines the semantic prototypes and visual features enabling the generator to synthesize reliable visual features. The authors design a visual-oriented semantic prototype evolving network to update the semantic prototypes iteratively. Experiments on three datasets show the effectiveness of the proposed approach.

**Summary Of The Review:**

This paper proposes a novel dynamic prototype network, which is a little complicated. Some errors may exist in the approach and experiments.

---

### Official Review · Reviewer_gkTe · 2022-10-25

**Confidence:** 4
**Correctness:** 3
**Technical Novelty And Significance:** 2
**Empirical Novelty And Significance:** 2
**Recommendation:** 5

**Clarity, Quality, Novelty And Reproducibility:**

The method of modifying a predetermined semantic prototype to an actual semantic prototype using visual features is recognized as a certain novelty.

On the other hand, the quality of the writing is low, making the paper difficult to read. For example, similar sentences appear many times. The content of the last paragraph of Section 1 is duplicated in the caption of Figure.2 and the content of the oveview in Section 3.

The conclusions of the experimental results are embedded in the description of the method (e.g., 1st and 2nd lines in page 5) and there is no reference to them.

Equation 2 calculates the L1 norm with $\hat{z}$ and $z_k$ where $\hat{z} \in \mathbb{R}^{|A|}$, but the definition of $z_{k+1}$ at the bottom of page 5 says $z \in \mathbb{R}^{C \times |A|}$.

Real and synthetic visual features are $x$ /  $\hat{x}$, but after mapping in V2SM, both features are $\hat{z}$, which is inconsistent in notation and confusing to the reader.

In section 3.1, there is no explanation for the difference between $z$, $\hat{z}$, and $z_{k}$.

In Figure 2, the input of the generator is the random variable $o$ not $x$ isn’t it?
It would help the reader's understanding if the meanings of the variables are also described in Figure 2.

There is no explanation for $x_{tr}$ and $x_{te}$.


**Strength And Weaknesses:**

- Strength

-- This paper proposes a new method for modifying predefined semantic prototypes to actual semantic prototypes using visual features.

-- Combining the proposed method with the ZSL method for a generative approach improves the performance of baselines methods.

- Weaknesses

-- The proposed method improves the performance of the generative ZSL methods, but is inferior to the embedding-based ZSL method.

-- The writing quality of the paper is low, making the paper difficult to read. The same explanations appear repeatedly in the paper. In addition, there are some typos.


**Summary Of The Paper:**

This paper proposes a dynamic semantic prototype learning method that aligns empirical semantic prototypes with actual semantic prototypes in order to synthesize accurate visual features. The method consists of a generative model that generates visual features from semantic prototypes, a network that maps visual features to semantic prototypes, and a network that modifies semantic prototypes by considering visual features. VOPE iteratively evolves predefined class semantic prototypes to become dynamic semantic prototypes. Using standard zero shot learning benchmark dataset, the proposed method improves the performance of the baseline generative approach ZSL methods.

**Summary Of The Review:**

Although the proposed method shows some novelty and performance improvement from the baseline, it is not considered to have reached the stage of publishing at this time due to its inferior performance compared to embedding-based methods and the low quality of writing.

---

### Official Review · Reviewer_LoBd · 2022-10-26

**Confidence:** 4
**Clarity, Quality, Novelty And Reproducibility:** Please see the model selection commen…
**Correctness:** 2
**Technical Novelty And Significance:** 3
**Empirical Novelty And Significance:** 3
**Recommendation:** 5

**Strength And Weaknesses:**

Pluses
- The method seems to be clearly defined, though in a confusing way (see below).
- Good experimental results: the experimental results show that the proposed method yields significant improvements based on multiple baselines.
- A somewhat detailed ablation study is also presented.
- Experimental results for various hyper-parameters are given.

Minuses
- While the method appears to be mostly well-defined, the paper does not give a clear and strong intuition why the method works (so well). The reasoning behind various modeling decisions are not fully clear. For example, it is not clear why z-tilde may really differ from z. In this sense, VOPE as a whole can also simply be seen as an extension of V2SM loss. Overall, despite understanding  Overall, I find it hard to understand the intuition behind the technical details of the model, i.e. there is a gap between the high level ideas mention in Section 1, and the way model is designed.
- As a continuation of my previous comment, I wonder whether VOPE could have been replaced by a simple class embedding-to-z_{k+1} mapping possibly with strong yet simple regularization (dropout, l2 regularization, etc).
The decisions in Section 3.4 are also not very clear. Why do you use dynamic prototypres for G conditioning? (Why conditioning do you use for G during training?)
- Similarly,  why do you do concatenation (Eq 9-10) ?  How does it affect the results?
- There is a lack of hyper-parameter tuning section since the application of invalid hyper-parameter tuning is a common malpractice in zero-shot learning studies. Could you please elaborate on your final model/classifier selection criteria? Do you perform the final model selection (together with loss weights, combination coefficient etc.) based on a validation set. Are the reported results on paper obtained without any further test set based finetuning?
- Minor: Performance evaluation of the proposed method on FLO dataset can be added since it is a widely used dataset among the works on generative ZSL and most of the state-of-the-art methods presented for comparison introduced FLO dataset in their evaluation scheme. (In this context: Figure 4 suggests that the details of certain HPs are highly sensitive.)


**Summary Of The Paper:**

The authors propose a semantic class embedding refinement mechanism by introducing two networks, (1) V2SM to enable the generator to synthesize semantically rich visual features and (2) VOPE to iteratively evolve the quality of semantic class embeddings throughout the training phase in generative zero-shot learning setting. Proposed refinement mechanism can be easily integrated into any existing generative ZSL pipeline just by simply introducing the proposed loss terms.


**Summary Of The Review:**

Overall, the experimental improvements are impressive. However, the paper does not give a strong intuition why the model works / why is it desiged as this. It might have been good to show each module's / step's effect on a simple (synthetic) 2D dataset example.

I have also concerns regarding the model selection procedure, given the sensitivity of the model to certain hyper-parameters.

---

### Official Review · Reviewer_qGfG · 2022-11-02

**Confidence:** 4
**Correctness:** 1
**Technical Novelty And Significance:** 1
**Empirical Novelty And Significance:** Not applicable
**Recommendation:** 3

**Clarity, Quality, Novelty And Reproducibility:**

It's hard to tell the clarity of the paper. It is wrong from the very beginning, but everything apart from that seems ok.

Novelty is out of the scope of discussion now as the paper has problems with correctness.

It would be impossible to reproduce the results as the experiments are in the wrong settings.

**Strength And Weaknesses:**

## Pros
---
* Good reference to the related articles
* The writing is easy to follow.

## Cons
---
This paper comes with tons of problems. I think there is no need to point out all of them because the design goes wrong from the very beginning.

Let us go directly to the Inference Phase of DSP. Training a softmax classifier in this paper requires a concatenated input of:
* The real/synthetic visual feature
* The evolved semantic prototype

How can one do classification by feeding a class prototype? Or let me ask how can one know which class prototype to concatenate before classifying the datum?

This design is breaching almost all the fundamentals of a recognition task. I must say the entire design is technically wrong.

Here are some notes on the other problems the authors need to be aware of:
* Eqs. (4) and (5) apply different distance measurements on the same feature space. Please consider making them consistent.
* Below Eq. (1), $z_{real} \cup z_{syn}=z\in R$ is totally a wrong expression.
* The model actually does not require momentum update (Eq. (6)). Initializing with the original semantic feature and then letting the gradient update z would be fine. Using Eq. (6) here is just a waste of time.

**Summary Of The Paper:**

This paper focuses on refining the semantic prototypes in ZSL. The authors top a V2SM network to the generator, mapping visual features to a semantic-related space. On the other hand, they transfer the semantic embeddings by another VOPE module. The outputs of V2SM and VOPE are aligned by cosine similarity and l2 losses. The most important step then is updating the semantic prototypes by the output of VOPE using momentum update.

The authors claim this procedure helps train ZSL classifiers and the experiments are conducted on conventional ZSL datasets.

**Summary Of The Review:**

As is discussed, this paper requires class labels as inputs for classification, which is a wrong design. I would reject this paper with no doubt. The authors might have made some mistakes in writing or it is just that they proposed something wrong. No matter what the cause is, this paper should not be considered for publication on any venue with its current form.

---

### Decision · Program_Chairs · 2023-01-20

**Decision:**

Reject

**Justification For Why Not Higher Score:**

Four knowledgeable reviewers recommend rejection. There is a rebuttal, and during the discussion, the reviewers reached a consensus that the paper has merits but is not ready for publication as there are several concerns in paper writings,  design choices that lacks sufficient justification, and missing experiments. The paper does not give a clear and strong intuition why the method work. No basis for overturning the reviews.

**Justification For Why Not Lower Score:**

N/A

**Metareview: Summary, Strengths And Weaknesses:**

The paper proposes a dynamic semantic prototype learning method by aligning empirical semantic prototypes with actual semantic prototypes to synthesize accurate visual features.  Although the ZSL task the authors are addressing is important,  there are several concerns (all reviewers recommended not to accept the paper in its current version). It is too long to go over them in this section, so I refer to the reviews below for details. In summary, there are several errors making the paper in its current version far from ready for publication.